# Enlarging the Scope of 5-Aminolevulinic Acid-Mediated Photodiagnosis towards Breast Cancers

**DOI:** 10.3390/ijms232314900

**Published:** 2022-11-28

**Authors:** Martin Kiening, Norbert Lange

**Affiliations:** Institute of Pharmaceutical Sciences of Western Switzerland, University of Geneva, Rue Michel-Servet 1, 1211 Geneva, Switzerland

**Keywords:** aminolevulinic acid, protoporphyrin IX, breast cancer, photodynamic diagnosis, theranostics

## Abstract

Today, most research on treating cancers targets one single cancer, often because of the very specific operation principle of the therapy. For instance, immunotherapies require the expression of a particular antigen, which might not be expressed in all cancers or in all patients. What about metastases? Combination therapies are promising but require treatment personalization and are an expensive approach that many health systems are not willing to pay for. Resection of cancerous tissues may be conducted beforehand. However, the precise location and removal of tumors are in most cases, hurdles that require margins to prevent recurrence. Herein, we further demonstrate the wide application of aminolevulinate-based photodynamic diagnosis and therapy toward breast cancers. By selecting four breast cancer cell lines that represent the main breast tumor subtypes, we investigated their ability to accumulate the fluorescent protoporphyrin IX upon treatment with the marketed 5-aminolevulinic acid hexyl ester (ALA-Hex) or our new and more stable derivative PSI-ALA-Hex. We found that all cell lines were able to accumulate PpIX under a few hours independent of their hormonal status with both treatments. Additionally, this accumulation was less dose-dependent with PSI-ALA-Hex and induced similar or higher fluorescence intensity than ALA-Hex in three out of four cell lines. The toxicity of the two molecules was not different up to 0.33 mM. However, PSI-ALA-Hex was more toxic at 1 mM, even though lower concentrations of PSI-ALA-Hex led to the same PpIX accumulation level. Additional illumination with blue light to induce cell death by generating reactive oxygen species was also considered. The treatments led to a dramatic death of the BT-474 cells under all conditions. In SK-BR-3 and MCF-7, ALA-Hex was also very efficient at all concentrations. However, increasing doses of PSI-ALA-Hex (0.33 and 1 mM) surprisingly led to a higher viability rate. In contrast, the triple-negative breast cancer cells MDA-MB-231 showed a higher death induction with higher concentrations of ALA-Hex or PSI-ALA-Hex. Derivatives of ALA seem promising as fluorescence-guided resection tools and may enable subsequent completion of cancer cell destruction by blue light irradiation.

## 1. Introduction

In 2020, breast cancer (BC) was the most commonly diagnosed cancer in women, with 2.3 million new cases (24.5% of female cancers), and the first cause of female cancer deaths (15.5%), with 685,000 new deaths [1]. X-ray-based mammography is considered the only effective screening method at this time [2], but recent research has revealed that this method is impaired by a high breast density and can be substituted by magnetic resonance imaging (MRI) [3]. Thus, research has been extending to other fields such as photodynamic diagnosis (PDD). The use of fluorescence in the detection of breast cancer has been recently reviewed by Grosenik and Bremer [4]. Most clinical trials have focused on the detection of sentimental lymph nodes during breast cancer surgery [5].

Whereas photodynamic therapy (PDT) targets a therapeutic effect by light destruction of cancer cells, PDD uses the fluorescence feature of the dedicated photosensitizer (PS) and its selectivity towards neoplasms to accurately spot the latter. Therefore, during fluorescence-guided surgery (FGS), the surgeon can perform several acts: (1) detect and localize cancerous fluorescent cells that cannot be observed with white light and confirm the status of visible abnormalities, (2) proceed to biopsies in the case of suspicion about the nature of the fluorescent tissue, and (3) resect the identified cancerous cells. Hence, 5-aminolevulinic acid (ALA)-mediated PDD is a potent resection tool that allows more precise removal of tumors, which prevents extensive injuries to healthy tissues by reducing the margin, while lowering the risk of recurrence by targeting additional lesions that cannot be seen with white light. This methodology has been recently approved for the fluorescence-guided resection of glioblastoma and has been generating a considerable interest towards many brain neoplasms [6]. Unfortunately, the ALA-induced fluorescence lacks sensitivity in various lesions, probably due to the low penetration ability of ALA. This problem was addressed by the hexyl ester of ALA (ALA-Hex), which is marketed under the trademarks Hexvix^®^ and Cysview^®^ for the improved detection of bladder cancer.

Several studies have demonstrated the feasibility of 5-ALA-mediated PDD in breast cancer and metastatic sentinel lymph nodes [7,8,9]. Other approaches, such as the aluminium phthalocyanine AlPcS_4_ or Hypericin for PDD and PDT of breast cancer have been examined [10,11]. Nonetheless, the former is hydrophilic, and its efficacy mainly depends on the lipid profile difference of cancerous and healthy tissues, while the latter is influenced by the variable expression of the breast cancer resistance protein (BCRP). Recently, the PDT effect of silicon phthalocyanine, ALA, rose Bengal and PpIX was also confirmed in ex vivo breast cancer tissues [12]. However, a clinical breakthrough in this area is still missing. Therefore, in this study, we investigated the ability of ALA-Hex and a new phosphatase-sensitive ALA derivative termed PSI-ALA-Hex [13] to induce in vitro protoporphyrin IX (PpIX) accumulation in breast cancer cells. Basically, this molecule comprises a hydrophilic ALA core, rendered more lipophilic with a hexyl-ester moiety added on the carboxylic acid side. As the obtained ALA-Hex molecule is unstable in solution and displays substantial toxicity, we stabilized it on the amine side with a “PSI” group, which stands for phospho-self-immolative, meaning that this precursor molecule will release the prodrug upon triggering by specific enzymes, phosphatases. The purpose is to keep the toxicity low subsequent to systemic administration, to increase bioconversion and to obtain higher therapeutic effects [14]. A main challenge when studying breast cancer is the high heterogeneity of the disease. In previous work, we already examined ALA-Hex and PSI-ALA-Hex in several cancer cell types including the breast MCF-7 [13,15]. To obtain a more comprehensive insight into the in vitro breast cancer response to PDD and PDT, we selected four breast cancer cell lines that represent the main breast tumor subtypes. These approaches could be used as a conventional tool for breast cancer independent of their hormonal status (Figure 1).

## 2. Results and Discussion

### 2.1. PpIX Induction

To assess the usability of ALA-Hex and PSI-ALA-Hex as breast cancer resection tools, we evaluated their capacity to induce fluorescent PpIX accumulation in vitro in four breast cancer cell lines expressing different hormonal statuses (Figure 2). The status is a crucial point when considering what is the best treatment, as the latter is often designed to target and block a specific receptor overexpressed in tumor cells, such as trastuzumab, an anti-HER2 antibody. Cell lines are labeled with “−” or “+” according to the expression level of estrogen receptors (ER), progesterone receptors (PR), and the human epidermal growth factor receptor 2 (HER2), in this order (ER/PR/HER2). BT-474 (+/+/+), SK-BR-3 (−/−+), MCF-7 (+/+/−), and MDA-MB-231 (−/−/−) cells constitute a group of triple positive or negative and HER2 positive or negative cells that provide a global insight into the comportment of both drugs in breast cancer.

The PpIX levels observed in Figure 2 displayed various time profiles depending on the cell line, the compound, and their concentrations. Upon ALA-Hex treatment, BT-474 and MDA-MB-231 produced a dose-dependent fluorescence observable in the very first hours of treatment. The heme biosynthesis pathway of BT-474 seemed to be saturated at 0.1 mM as higher ALA-Hex concentrations led to reduced fluorescence over time. This phenomenon might be due to the lower resistance of the BT-474 cells to the treatment with ALA-Hex. At 24 h, the 0.33 mM and 1 mM-induced fluorescence intensities were reduced by 20% and 73%, respectively. In contrast, in MDA-MB-231, the higher the concentration, the higher the fluorescence and the later the peak. As a matter of fact, from 0.033 mM to 1 mM, the maximum fluorescence was reached at 2 h, 4 h, 6 h, and 24 h, and the subsequent PpIX elimination was delayed accordingly. In SK-BR-3 (up to 12 h) and MCF-7 (up to 30 h), no difference in fluorescence production rates was noticed from one concentration to another (*p*-Values > 0.05), except for 1 mM ALA-Hex in MCF-7 whose fluorescence increased for 2 h up to a low level, to subsequently remain stable over the 30-h monitoring. This suggests that the PpIX production was tightly regulated by a limiting step with treatment as low as 0.033 mM in SK-BR-3 and MCF-7. 

Interestingly, treatments with equivalent molar concentrations of the ALA-Hex phosphate derivative, PSI-ALA-Hex, exhibited more even profiles than ALA-Hex. Indeed, in BT-474, a significant difference was only observed at 30 h between the 0.33 mM and 1 mM treatments (*p*-Value = 0.03), the latter showing a 27% lower fluorescence. In SK-BR-3, significant differences were observed at 24 h between 0.033 mM and 1 mM PSI-ALA-Hex (*p*-Value = 0.02), and 0.1 mM versus 1 mM (*p*-Value = 0.01) with a fluorescence intensity reduced by 11% (0.033 mM) and 12% (0.1 mM) compared to the 1 mM treatment. The heterogeneous profile obtained after 12h with ALA-Hex was not reproduced with PSI-ALA-Hex. We hypothesize that the increased stability of our new derivative allowed it a longer release.

MCF-7 cells treated with PSI-ALA-Hex produced a significantly lower fluorescence at 24 h with 0.33 mM (*p*-Value = 0.03) and 1 mM (*p*-Value = 0.001) compared with the 0.033 mM treatment. However, contrary to the 1 mM ALA-Hex, which induced an 86% lower PpIX accumulation compared to 0.33 mM, the 1 mM PSI-ALA-Hex performed better with a minor fluorescence reduction of 29% compared to 0.033 mM. Curiously, the MDA-MB-231 cells did not accumulate as much PpIX with PSI-ALA-Hex compared with ALA-Hex. For example, 1 mM PSI-ALA-Hex led to 47% less fluorescence than 1 mM ALA-Hex at 24 h (*p*-Value < 0.001). Surprisingly, the PpIX accumulation rate was similar between the 0.33 mM and 1 mM PSI-ALA-Hex up to 12 h (*p*-Value = 0.06), suggesting a saturation of the anabolism, while 1 mM ALA-Hex induced significantly higher fluorescence than 0.33 mM in as little as 4 h (*p*-Value = 0.007).

Altogether, these results suggest that both drugs are suitable for PDD in breast cancer, independent of their hormonal status. We observed a lower dose-dependency using PSI-ALA-Hex instead of ALA-Hex in three out of four cell lines while reaching fluorescence intensities as high as those obtained with ALA-Hex. Although the level of PpIX produced in MDA-MB-231 seems limited with the PSI-ALA-Hex treatment, a 0.33 mM concentration induced a fluorescence level similar to the ALA-Hex results.

### 2.2. Dark Toxicity

We investigated whether 5-ALA derivatives elicited toxicity towards the panel of analyzed cells, which could explain the lower PpIX accumulation observed with some of the highest treatment concentrations. Figure 3 presents the viability of the four breast cancer cell lines treated with identical concentrations of ALA-Hex or PSI-ALA-Hex and incubated in the dark for 30 h.

BT-474 and SK-BR-3 exhibited similar profiles—a slight viability reduction with increasing concentrations, but similar cell viability between both molecules at a given concentration, even though in SK-BR-3, the PSI-ALA-Hex was significantly less toxic than ALA-Hex at 0.1 and 0.33 mM. Noticeably, the only substantial difference was observed for BT-474 in which a 1 mM PSI-ALA-Hex treatment drastically dropped the viability to 9% compared to 1 mM ALA-Hex (58%) (*p*-Value < 0.001), which might in turn explain the reduced PpIX production at this concentration. In the MCF-7 cells, no toxicity was observed with either of the ALA-Hex concentrations or the PSI-ALA-Hex up to 0.33 mM. However, 1 mM PSI-ALA-Hex lessened the MCF-7 viability to 56% (*p*-Value < 0.001), where 0.33 mM and 1 mM ALA-Hex significantly enhanced it to 122% (*p*-Value = 0.04) and 135% (*p*-Value = 0.001), respectively, compared to the untreated cells. This rise was unexpected and only observed in MCF-7. Finally, MDA-MB-231 behaved in a third way, which was the lack of ALA-Hex toxicity at any concentration, but a mild PSI-ALA-Hex toxicity at 0.33 mM (82% viability) (*p*-Value = 0.007) and strong toxicity at 1 mM (15% survival) (*p*-Value < 0.001) compared to the control. Globally, ALA-Hex and PSI-ALA-Hex induced similar toxicities in all cell lines from 0.033 to 0.33 mM, while the 1 mM PSI-ALA-Hex concentration led to a significant viability drop in three out of four cell lines.

### 2.3. Photodynamic Therapy

As shown in Figure 4, we assessed the ability of blue light to induce cell death by activation of the PpIX photosensitizer and generation of toxic reactive oxygen species (ROS). We used a range of fluences of 5, 10, and 20 J/cm^2^ (Figure 4, Appendix A) to evaluate the PDT potential of our molecules in all cell lines, even at low concentrations. This choice was based on previous studies in bladder, lung, or prostate cancer cells [13,16,17] and considering that in vivo protocols usually choose fluences from 20 to 100 J/cm^2^ [18,19]. In BT-474, all conditions resulted in dramatic cell death. SK-BR-3 and MCF-7 reached almost integral toxicity by ALA-Hex treatment at all concentrations, while PSI-ALA-Hex-induced toxicity was inversely proportional to the dose, as observed at 0.33 mM and 1 mM (up to 71% viability for SK-BR-3). This suggests that a lower amount of photosensitizer was previously produced, which led to limited ROS generation and toxicity. However, this hypothesis seems ambiguous when comparing the PpIX levels and the light toxicity. Indeed, in SK-BR-3, 1 mM PSI-ALA-Hex showed a fivefold lower toxicity than 0.033 mM (Figure 4), when their fluorescence profiles (Figure 2) were superimposed.

In contrast, the triple-negative MDA-MB-231 cells treated with 0.033 mM and 0.1 mM concentrations of either molecule were more resistant to 5 J/cm^2^ PDT (Figure 4). Although ALA-Hex induced absolute toxicity from 0.33 mM, increasing concentrations of PSI-ALA-Hex from 0.1 mM did not enhance the PDT efficacy. However, its fluorescence profile (Figure 2) showed a higher PpIX generation in the 0.33 mM and 1 mM treatments compared with 0.1 mM at all the time points. When comparing both drugs’ PpIX profiles, we can assume that the higher accumulation of PpIX induced by ALA-Hex was responsible for the better response to PDT.

As shown in the Appendix A, the use of higher fluences substantially increased the PDT efficacy of PSI-ALA-Hex in MCF-7 by changing the 1 mM cell viability from 38% at 5 J/cm^2^ (Figure 4) to 9% at 20 J/cm^2^ (Appendix A). Even though BT-474 and SK-BR-3 did not undergo a notable improvement with the light dose, the MDA-MB-231 cells were further affected at 10 J/cm^2^ (Appendix A) and thoroughly destroyed at 0.1 mM, 0.33 mM, and 1 mM of both molecules when irradiated at 20 J/cm^2^ (Appendix A). Surprisingly, at the lowest concentration of 0.033 mM, neither ALA-Hex nor PSI-ALA-Hex displayed a significant improvement in PDT. This feature suggests that a minimum PpIX level is necessary to induce toxicity, independent of the light fluence. The cellular localization of PpIX was already suggested to be a critical parameter for efficient ROS generation and PDT efficacy, even though it does not seem to be the main responsible aspect in this case.

The data presented in this study, focused on breast cancer, confirmed the results of our previous study conducted on cancer cells of prostate, bladder, and lung cancer origin [13]. Most importantly, PSI-ALA-Hex seems to be suitable to detect and treat breast cancer regardless of the hormonal status. In a preliminary study, Millon et al. investigated in vitro the use of ALA in breast cell lines including MCF-7 and MDA-MB-231 [20]. They were able to discriminate normal from cancerous cells by spectroscopy after treatment with 3 mM ALA. More recently, a study from Eskiler et al. suggested that MDA-MB-231 cells were more sensitive to 1 mM ALA-mediated PDT (at 9 and 12 J/cm^2^) [21].

In a previous phase II trial, Ottolino-Perry et al. demonstrated the feasibility of using ALA for the detection of breast cancer. However, due to the lower stability and low bioavailability of ALA, the search for alternative compounds suitable for systemic administration is still a valid research area. While our results using 1 mM ALA hexylester demonstrated a complete elimination of both cell types at 5 J/cm^2^, which did not allow us to conclude on the cell sensitivity to blue light, we clearly saw that lower concentrations (0.033 and 0.1 mM) were more efficient on MCF-7 than MDA-MB-231 (Figure 4). For both alternative treatments, the reduction in cell viability is in accordance with the generation of PpIX (cf. Figure 2 vs. Figure 4). However, due to the low therapeutic index of ALA-Hex and the lower systemic toxicity of PSI-ALA-Hex [15], the latter is a suitable candidate to pursue research on more relevant models including three dimensional or in vivo models.

## 3. Materials and Methods

### 3.1. Cell Culture

Human breast ductal carcinoma cell line BT-474 (American Type Culture Collection (ATCC^®^) HTB-20™, Manassas, VA, USA) and human breast gland adenocarcinoma cell lines SK-BR-3 (ATCC^®^ HTB-30™, Manassas, VA, USA), MCF-7 (AddexBio C0006008, San Diego, CA, USA), and MDA-MB-231 (AddexBio C0006002, San Diego, CA, USA) were cultured as monolayers in vitro. They were fed with media constituted of Hybri-Care (ATCC^®^ 46-X™) + 0.15% NaHCO_3_ (BT-474), McCoy’s 5A (ATCC^®^ 30-2007™) (SK-BR-3), MEM (Gibco™ 41090028) (MCF-7), and DMEM 4.5 g/L glucose (Gibco™ 31966021) (MDA-MB-231). All media were supplemented with 10% fetal calf serum (CVFSVF00-01, Eurobio, Courtaboeuf, France) and 1% penicillin–streptomycin mix (Gibco™ 15140122). Cells were cultivated at 37 °C under 95% air and 5% CO_2_ humidified atmosphere. They were routinely passaged using 0.25% trypsin–EDTA mix (Gibco™ 25200056) and regularly checked for mycoplasma contamination with the MycoAlert™ detection kit (Lonza LT07-218, Basel, Switzerland).

### 3.2. PpIX Induction

Cells were lifted with trypsin–EDTA, stained with ViaStain™ AOPI (acridine orange/propidium iodide) solution (Nexcelom Bioscience, Lawrence, KS, USA), and counted with the Cellometer Auto 2000 automated cell counter (Nexcelom Bioscience, Lawrence, KS, USA). BT-474 and SK-BR-3 were seeded at 40,000 cells/well, and MCF-7 and MDA-MB-231, at 10,000 cells/well in 96-well black plates (clear bottom, 3603, Costar^®^, Corning Incorporated, Kennebunk, ME, USA). Upon reaching confluency, the cells were washed with 200 µL of DPBS (Gibco™ 14190094) per well and treated with 100 µL of 0.033, 0.1, 0.33, or 1 mM of ALA-Hex or PSI-ALA-Hex [15] freshly prepared in an FCS-depleted medium. Induction of fluorescent PpIX was recorded with a Safire microplate reader (Tecan, Männedorf, Switzerland) at several time points (0 h, 1 h, 2 h, 4 h, 6 h, 8 h, 10 h, 12 h, 24 h, and 30 h post-treatment) using the following parameters: preheating at 37 °C, λexc = 405 ± 12 nm, λem = 635 ± 12 nm, gain = 120, integration time = 40 µs, and multiple reads per well 3 × 3. The mean values (n = 3 or 4) and SD of the treated conditions were normalized by subtracting the blank (cells with FCS-depleted medium). Statistical analysis was performed by two-way ANOVA (uncorrected Fisher’s LSD).

### 3.3. Dark and Photo Toxicities

Cells were lifted with TrypLE™ (Gibco™ 12604021), stained with trypan blue, and counted with the Countess™ 3 automated cell counter (Invitrogen™, Life Technologies Holdings Pte Ltd., Singapore). Subsequently, 20,000 cells/well were seeded in 100 µL of the medium in 96-well transparent plates (clear bottom, 3595, Costar^®^, Corning Incorporated, Kennebunk, ME, USA) that were incubated at 37 °C, 5% CO_2_. After 24 h, cells were treated with 100 µL of freshly prepared ALA-Hex or PSI-ALA-Hex twice concentrated in medium to reach 0.033 mM, 0.1 mM, 0.33 mM, or 1 mM of the corresponding drug. “Live” control wells were completed with the medium, and “dead” control wells were completed with SDS (0.25% final). After 6 h of treatment, cells were either kept in the dark or irradiated with a LumiSource^®^ illuminator (PCI Biotech AS, Oslo, Norway) equipped with 4 blue light tubes (Osram L 18/67, 450 nm, 8.92 mW/cm^2^) at a blue light fluence of 5 J/cm^2^, 10 J/cm^2^, or 20 J/cm^2^. The toxicity of treatments was evaluated 24 h post-irradiation with the colorimetric WST-1 assay (Roche 11644807001). Cells were washed once with 200 µL of DPBS and incubated with 100 µL of medium/WST-1 (10:1) solution for 1.5 h (MDA-MB-231 and BT-474), 2 h (MCF-7), or 3 h (SK-BR-3) to reach a sufficient formazan coloration for the “live” control. Absorbances were measured with a Safire microplate reader (Tecan, Männedorf, Switzerland) using the following parameters: preheating at 37 °C, initial orbital shaking: normal strength 60 s, λabs = 450 nm, λref = 690 nm, and 10 flashes per well. The cell survival was calculated using the following formula:(1)$${\%}_{alive}=100\times \frac{A_{i}-A_{✕}}{A_{✓}-A_{✕}}$$
where %*_alive_* is the percentage of cell survival, *A_i_* is the absorbance of the tested condition, *A*_✓_ is the absorbance of the live control, and *A*_✕_ is the absorbance of the dead control. The values are expressed as the mean absorbance ± standard deviation.

### 3.4. Software

The figures and statistics were generated using GraphPad Prism version 9.1.2 for Windows (La Jolla, CA, USA). 

Figure 1 was created with BioRender.com, accessed on 11 May 2022.

## 4. Conclusions

PSI-ALA-Hex, a more stable derivative of the marketed photosensitizer prodrug aminolevulinate hexylester, was tested along with ALA-Hex in a panel of breast cancer cell lines to assess their ability to induce the fluorescent protoporphyrin IX. The photodynamic diagnosis was evaluated along with the PDT potential. Ideally, administration of the ALA derivative should induce a selective accumulation of PpIX in breast cancer cells under a few hours to help surgeons removing cancer cells accurately by fluorescence-guided surgery, and ideally, processing to the photodestruction of the potential remaining cells.

Our study demonstrates the ability of both drugs to induce PpIX in four breast cancer cells, independent of their hormonal status, that is, the expression of receptors usually targeted in immunotherapies. The response to PSI-ALA-Hex was less dose-dependent than ALA-Hex in three out of four cell lines, which suggests the possibility of lowering the administrated dose while keeping an equal detection level and limiting the acute toxicity. Both drugs showed a limited dark toxicity up to 0.33 mM. Triple-negative MDA-MB-231 cells were sensitive to the two derivatives, with a preference for ALA-Hex at 1 mM. However, this concentration showed the highest dark toxicity and did not necessarily improve the PDT effect.

To conclude, both drugs can be used to detect and kill these breast cancer cell lines in vitro depending on the concentration and the fluence that is used. The choice of the molecule is important, as the higher stability of PSI-ALA-Hex could ease the preparation of the treatment ahead of the administration that includes the reconstitution process in a sterile way that might be delicate. Additionally, these drugs may not induce breast cancer resistance to PDT, a feature that should be investigated to help choosing the best candidate.

## Figures and Tables

**Figure 1 ijms-23-14900-f001:**
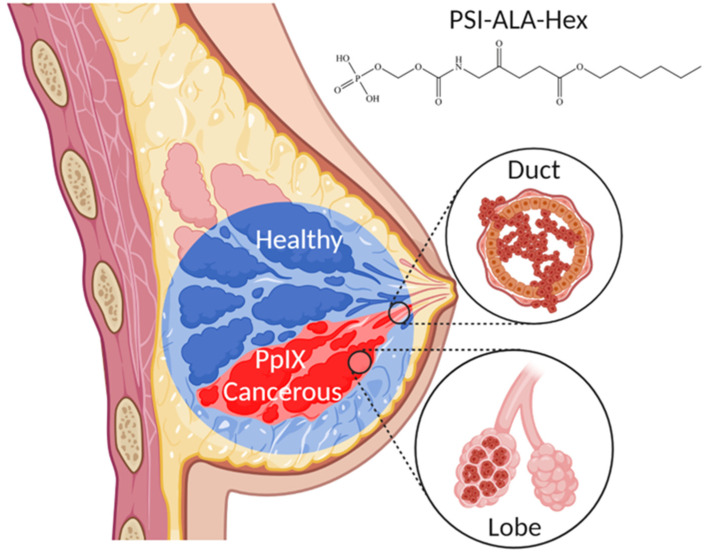
Conceptualization of breast cancer photodynamic diagnosis by selective accumulation of PSI-ALA-Hex-induced protoporphyrin IX and red fluorescence detection with blue light.

**Figure 2 ijms-23-14900-f002:**
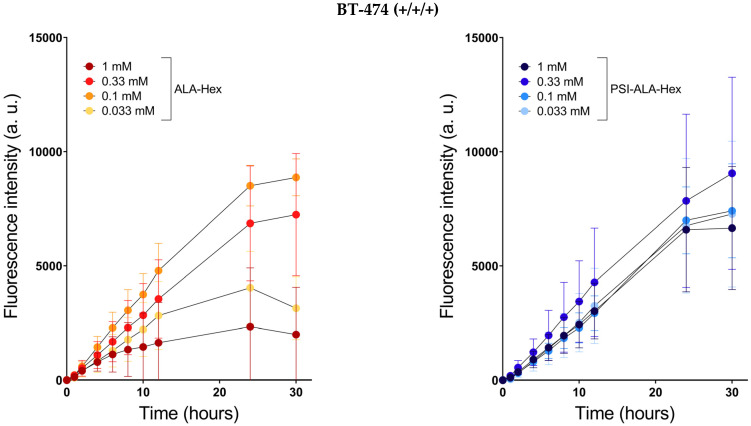
Protoporphyrin IX induction capacity (fluorescence intensity) of ALA-Hex (orange shades) and PSI-ALA-Hex (blue shades) at concentrations of 0.033 (•/•), 0.1 (•/•), 0.33 (•/•), and 1 mM (•/•) in BT-474, SK-BR-3, MCF-7, and MDA-MB-231 breast cancer cells in vitro.

**Figure 3 ijms-23-14900-f003:**
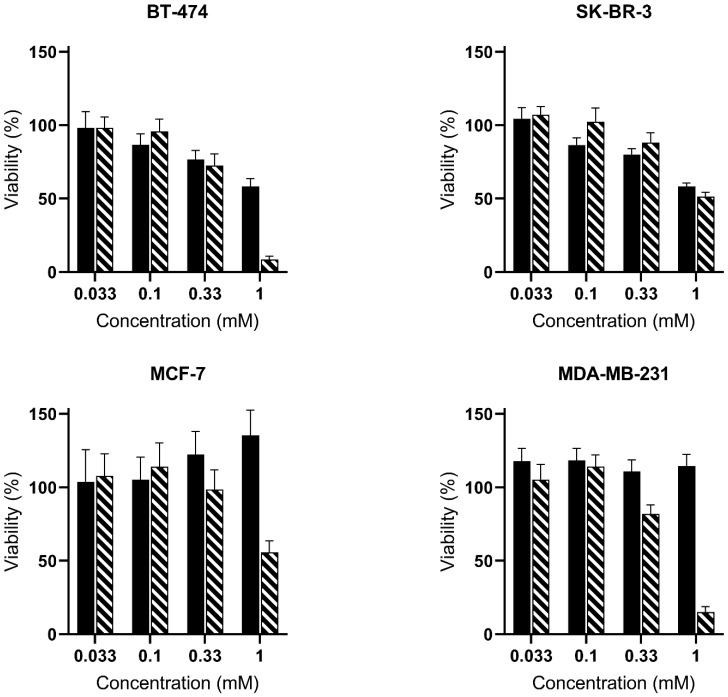
Evaluation of the dark toxicity of ALA-Hex (
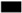
) and PSI-ALA-Hex (
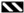
) at concentrations of 0.033 mM, 0.1 mM, 0.33 mM, and 1 mM after 30 h of incubation by WST-1 viability measurement in BT-474, SK-BR-3, MCF-7, and MDA-MB-231 breast cancer cells in vitro.

**Figure 4 ijms-23-14900-f004:**
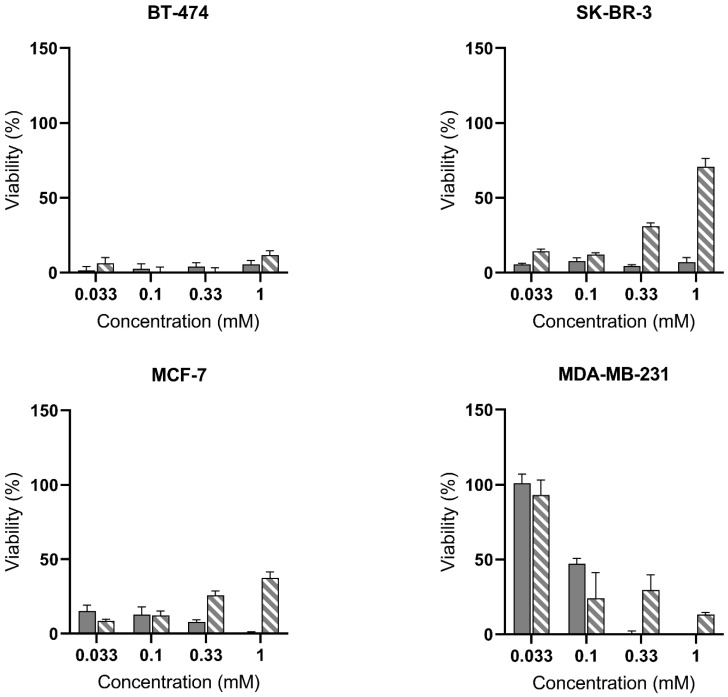
Evaluation of the photodynamic therapy potential of ALA-Hex (
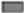
) and PSI-ALA-Hex (
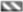
) at concentrations of 0.033 mM, 0.1 mM, 0.33 mM, and 1 mM in BT-474, SK-BR-3, MCF-7, and MDA-MB-231 breast cancer cells in vitro. Cells were incubated for 6 h then exposed to blue light (5 J/cm^2^), and the PDT efficacy was assessed 24 h post-irradiation by WST-1 viability measurement. Values are normalized against the untreated control exposed to the same blue light fluence.

## Data Availability

Not applicable.

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
