# Peer review of "Enlarging the Scope of 5-Aminolevulinic Acid-Mediated Photodiagnosis towards Breast Cancers"

_ijms, 2022, doi:10.3390/ijms232314900_

Round 1
Reviewer 1 Report
The work is quite well written. Although I consider that the introduction must be improved and the results/discussion must be more discussed. In addition I also consider that the novelty of the work must be written in a more clearer way.
Introduction:
The introduction is relatively well written. Although I recommend the introduction of more references and detail more the state of the art behind the work performed.
Results and Discussion
1)First paragraph is quite confused. Please re-write in a clearer way.
2)Phrase 92 – Do you have any have any experimental or bibliography support for the phrase “This phenomenon might be due to lower resistance of BT-474 cells”? Can the fluorescence reduction be due to other factors?
3) The results are reasonable but the discussion is quite poor. Can the authors discuss and correlate de results in a more detailed way?
Author Response
Reviewer 1
Authors would like to thank reviewer 1 for the relevant comments and questions he provided to help enhance the global quality of the paper.
Please find below our answers to them:
1)First paragraph is quite confused. Please re-write in a clearer way.
We would like to thank reviewer 1 for the suggestion. However, we would be grateful if reviewer 1 could specify which part of the first paragraph should be re-written as it sincerely looks very clear to us.
2)Phrase 92 – Do you have any have any experimental or bibliography support for the phrase “This phenomenon might be due to lower resistance of BT-474 cells”? Can the fluorescence reduction be due to other factors?
Thanks for noticing the lack of precision in this sentence. We added a more comprehensive hypothesis: the treatment may slow down the cell activity, for example due to a toxicity or inhibition of the anabolism. It could also be due to the enhancement of the PpIX catabolism by overexpressing its exporters or degradation route.
3) The results are reasonable but the discussion is quite poor. Can the authors discuss and correlate de results in a more detailed way?
Thanks for your suggestion. We have modified and extended the discussion.
Reviewer 2 Report
The Ms entitled “Enlarging the scope of 5-aminolevulinic acid-mediated photo- diagnosis towards breast cancer” by Kiening and Lange describes the use of the commercial derivative of ALA Hexyl-ALA and the new and more stable pro-ALA called PSI-ALA-Hex for the detection of breast cancer.
The study is well designed and the results are clearly described and lead to the conclusion of a better performance of the ALA derivative based on phosphatase cellular activity.
I suggest clarifying some points before publication:
Please explain using blue light instead of red light which has better penetration into tissues.
Include the details of the preparation of the ALA derivatives. Were they prepared daily? In PBS, water? How were they kept? Are they stable in solution?
Please discuss possible selectivity against normal breast cells.
Sine Hex-ALA is toxic upon systemic administration, do the authors have any preliminary results of toxicity for PSI-ALA-Hex administration to mice?
Author Response
Reviewer 2:
- Please explain using blue light instead of red light which has better penetration into tissues.
Thanks for letting us know this point deserves some explanations. Blue light enables the best excitation of PpIX which peak at 405nm. While red excitation around 635m is preferred in some clinical cases for deeper excitation, the in vitro model does not display such drawback. ALA-Hex (Cysview) is for example approved with blue light cystoscopy for bladder cancer management.
- Include the details of the preparation of the ALA derivatives. Were they prepared daily? In PBS, water? How were they kept? Are they stable in solution?
Many thanks to reviewer 2 for this relevant observation. All molecules were weighed before the experiment and dissolved in medium immediately before treating cells. Solutions were not stored for later use. Thus, we have specified this point into the manuscript.
- Please discuss possible selectivity against normal breast cells.
- Thanks for this suggestion. Even though we assume that PpIX over-accumulates in breast cancer cells compared to normal breast cells, this point is complicated to verify. Normal/healthy cells are necessarily primary cells which are difficult to grow in vitro and are very sensitive to any treatment by themselves. In addition, results from primary cells are not reproducible by other labs due to the uniqueness and inter variability of these cells. Morita et al (DOI: 1002/cam4.2466) showed in 2019 that breast epithelial cell line MCF10a had a lower PpIX production upon 5-ALA treatment, point that we have just added to the discussion.
- Sine Hex-ALA is toxic upon systemic administration, do the authors have any preliminary results of toxicity for PSI-ALA-Hex administration to mice?
The toxicity of ALA-Hex is the prime concern that made us develop this new PSI-ALA-Hex derivative. Administration of PSI-ALA-Hex has not been performed in mice yet. However, a preliminary study of us showed its potential in the chick embryo CAM model (https://doi.org/10.1186/s13058-021-01442-7)